# Data Quality—Concepts and Problems

**Max J. Hassenstein \*,† and Patrizio Vanella ‡**

Department of Epidemiology, Helmholtz Centre for Infection Research (HZI), 38124 Braunschweig, Germany; patrizio.vanella@helmholtz-hzi.de

\* Correspondence: max.hassenstein@helmholtz-hzi.de

† PhD Programme "Epidemiology" Braunschweig-Hannover, Germany.

‡ Chair of Empirical Methods in Social Science and Demography, University of Rostock, 18057 Rostock, Germany.

**Definition:** Data Quality is, in essence, understood as the degree to which the data of interest satisfies the requirements, is free of flaws, and is suited for the intended purpose. Data Quality is usually measured utilizing several criteria, which may differ in terms of assigned importance, depending on, e.g., the data at hand, stakeholders, or the intended use.

**Keywords:** data quality; information quality; data quality dimensions; data life cycle





## 1. Introduction—History, Disambiguation and Scope

The word data is the plural form of the Latin noun datum (verbatim "something given") [1]. In general, data is "information, especially facts or numbers, collected to be examined and considered and used to help decision-making, or information in an electronic form that can be stored and used by a computer" [2]. The sense of data may vary depending on the context; for example, researchers often work with data sets, which are data in an accumulated and structured, often in tabularized form.

Old papyrus fragments from ancient Egypt (specifically, from the 26th century BC) indicate the early existence of logbooks and documentation, thus proving data collection as a phenomenon as old as early civilizations [3]. The Roman Empire, for instance, also recorded and collected data as evidenced by its historical censuses to create population registers containing asset estimations and medical examinations for military service [4].

Today, and due to the digital age, data have become omnipresent in private, commercial, political and scientific environments. Computing underwent drastic transformation within the past 40 years: until the 1980s, centralized data centers gathered data and were business-orientated, and by 2000, data centers expanded their data management capabilities, and individual users increasingly had access to a private computer and the World Wide Web (WWW) [5]. Since 2000 and with the increasing spread of the internet, data centers have expanded their capacities to cloud computing, resulting in considerably increased amounts of data collected and available [5].

Shannon [6], a pioneer of information theory, defined information as a simple unit of message (e.g., a binary digit, known as bit), either stand-alone or as a sequence, sent by a sender to a receiver. However, we see a certain degree of distinction between the terms data and information; from our point of view, data are relatively solitary and of a technical nature, and require interpretation or placement to become information [7,8].

The word quality has multiple origins, among others, from the Latin noun qualitas (verbatim characteristic, nature). According to ISO 9000:2015 [9], quality is the "degree to which a set of inherent characteristics of an object fulfills requirements." Nevertheless, the requirements remain undefined at this point. Therefore, in our context, quality broadly refers to the extent of the goodness of a thing (for instance, our data).

Based on the presented terms for quality and data, a definition for data quality can already be deduced: the degree to which the data of interest fulfills given requirements, as

similarly defined by Olson [10]. However, the literature offers additional interpretations of the data quality concept. These are, in essence: Whether the data are fit for (the intended) use and free of flaws [11] or meet the needs and requirements of their users [12,13]. In this regard, data quality requirements may be imposed by standards, legislation, regulations, policies, stakeholders, or their intended use [14].

For instance, the wide availability of modern information technology, such as smart devices (phones, tablets, and wearables), has made people eager to track their physical activity, sleep and other health data, or dietary habits as a hobby [15,16]. Likewise, companies have turned data into a business model (for instance, Google or Meta, previously known as Facebook) or accumulate data for knowledge management. Furthermore, specific scientific disciplines, such as epidemiology, acquire data to research health conditions and their causes [17]. These are just a few examples of how much data has become part of everyday life. However, the ubiquity of data goes hand in hand with the ubiquity of data quality issues. Simple examples from everyday life are outdated phone numbers or unregistered changes of residence in a contact directory, which may lead to an inability to contact a particular person or bias statistical analyses that consider geographical variables, challenging the usefulness of the directory.

The quality and applicability of data should not and cannot be assumed by default, as they may directly impact data processing as well as the results and conclusions derived from the data [18,19].

High-quality research and analyses require reliable data, frequently referenced inversely as "garbage in, garbage out" [20,21]. Even if, from our point of view, quality considerations concerning the data collected might be as old as the collection procedure itself, we only find the rather modern literature to discuss this matter [22,23]. Nevertheless, data quality was already labeled "a key issue of our time" [24], at a much lower digitization level in 1986.

The primary motivation for work in the field of data quality is generally to ensure data integrity and, thus, in principle, to ensure the usability and usefulness of the data. Thereby, the stakeholders of data quality are data producers, data users, analysts, and people who derive conclusions from interpreted data, such as the readership of a study or the recipients of information provided via the WWW. Regardless, data quality considerations should primarily concern the people either involved in data collection, data generation, or those analyzing or providing data, as well as people with direct data access, as they have the means to address data quality issues. Ideally, data quality considerations precede and accompany the data collection phase and may imply, for example, measures to assure the data structure or value range controls. However, as discussed in Section 2.2.3, quality assurance may be a continuous process.

Our contribution is structured as follows. The following section presents data quality concepts and discusses data quality assessment within these frameworks. Section 3 illustrates data quality issues in real-life examples, focusing on the health sciences, to give the readers a better grasp of the theoretical concepts presented earlier. Finally, Section 3.3 describes the challenges associated with the practical application of the data quality frameworks before we close the paper with a conclusion in Section 4.

## 2. Data Quality: Context and Measurement

### 2.1. Data Life Cycle

A life cycle model may be used to illustrate the phases or developments of a project, product, or construct in an overall context. Generally, there are project life cycles and system life cycles; while the project life cycle ends at the end of the project, the system life cycle result in a loop and only end upon system disposal [25]; we assign the data life cycle (Figure 1) to this category. The cycle may consist of two general sections: a data reuse phase (e.g., the use of open data [26], data that can be used, processed, and shared by anyone for any purpose) and the project phase. The data description and provision steps serve to complete the cycle. In the data reuse phase, data that had previously been generated and

made available is reused; during the process, it may be extended or reduced to ultimately generate new data. Suppose required data is not available in the data life cycle. In that case, the data may be created and incorporated into the cycle during a stand-alone, traditional project, often based on findings from previous work and data. Especially, in the context of open data and their wide availability, the illustration of the data life cycle may serve to identify the possible sources of data quality problems and the starting points for data quality assurance.

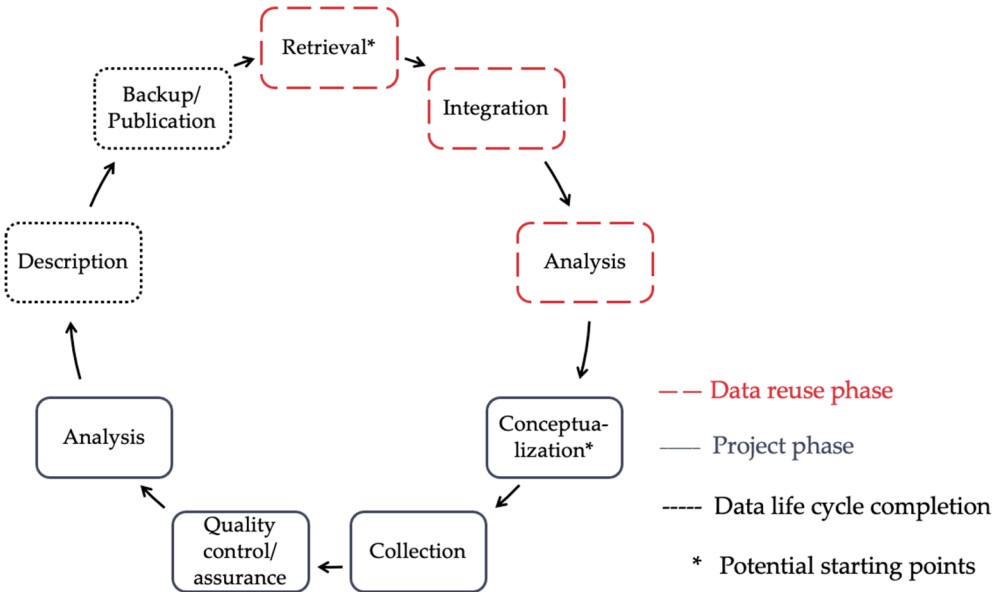

**Figure 1.** The data life cycle, suggested by Rüegg et al. [27], own illustration.

Even if quality control and quality assurance have their own node in the data life cycle, data quality investigations or at least considerations about it are practically feasible at every other node, as the individual points of the data life cycle are susceptible to all kinds of potential challenges and problems, which are addressed exemplarily in Section 3.3.

In addition to the stand-alone node for data quality management within the data life cycle, several activities may be undertaken during the data acquisition or usage phases to diminish the consequences of poor data quality, counteracting or even improving them, which a data management plan [28] can specify. These activities may involve constant data quality monitoring to identify issues and data quality assessment reports for communication and transparency regarding the state of the data. Also, data cleansing activities allow for a prompt response to quality issues or to potentially enable improvements during the data collection process, for example, by establishing data input control measures in digital questionnaires (see Section 3.2 for a case study). However, the individual activities must be coordinated with each other in comprehensive data quality management.

*2.2. Demands, Dimensions and Approach*

2.2.1. The FAIR Principles and Good Scientific Practice

Sustainable research data management requires the (meta-) data of interest to be archived in a way to enable data reuse, implying data quality requirements. In this context, Wilkinson et al. [29] suggested the FAIR Guiding Principles for scientific data management and stewardship—standing for Findable, Accessible, Interoperable, and Reusable—which should ensure the sustainability of research data. In summary, the FAIR principles state the following:

- Data are findable when they have a unique identifier, comprehensive metadata are provided, and if both the data and the metadata are indexed or registered to provide discoverability.

- Data are accessible when both the data and the metadata are retrievable, using a unique identifier and a free and open standardized communication protocol (while being generally implementable and providing features for authentication or authorization). Furthermore, the metadata availability should exceed that of the original data.
- For interoperability, the data and metadata must adhere to a standard form and be provided in a universal language. Both the data and the metadata should refer to other existing data.
- Regarding reusability, both the data and the metadata must be depicted in detail, including the statement of usage licenses along with their publication and the exact explanations of the data origin, and should generally be shared following standards used in the particular domains.

Thus, from our point of view, immediate requirements arise for both the data and its quality during the sharing process, especially for the Accessibility dimension, a measure of data accessibility (see Section 2.2.2). In practical applications, the FAIR principles have been shown to improve data quality in data provision of European marine litter data [30].

Furthermore, the guideline of the German Research Foundation (Deutsche Forschungsgemeinschaft, DFG) to ensure good scientific practice [31] explicitly demands measures for continuous quality assurance, which should accompany the overall research process. Therefore, the quality of research data must be addressed from our point of view, providing further motivation to deal with the issue of data quality. The Council for Information Infrastructures at the Karlsruhe Institute for Technology (KIT) additionally presents five general concepts for the (self-) control of scientific actions in the context of research data quality [32], namely:

1. a control mode of juridical nature introducing norms and standards (for example, professional or ISO-norms),
2. the organizational nature relying on incentives and enhancements using data validation and organizational or procedure-based operationalizations of data quality (certificates, quality seals),
3. contractual and communication-oriented approach-setting policies and guidelines for the handling of research data and its data management plans,
4. the procedural ideal for the assurance of data quality following (for example) data life cycles (idealistic and schematic descriptions of processes to be improved, see Section 2.1), and
5. the pragmatic and procedural control of quality developments specifying generalistic formula as fit-for-use, or the FAIR principles.

### 2.2.2. Data Quality Dimensions

Diversity in data use and data purposes may result in different requirements for data quality [12]. Here, universal and objective measures for determining data quality play a significant role and add valuable criteria to somewhat subjective criteria such as the relevancy of the data to the user.

The concept of the so-called data quality dimensions aims to grasp the underlying ideas of data quality, in which each dimension represents a specific characteristic. Wang and Strong [12] were early advocates and established the dimensions from a fit-for-use data user perspective in 1996. They identified data quality attributes that consolidate in 20 dimensions grouped into four overarching data quality categories: Intrinsic, contextual, representational and accessible (Figure 2). Here, intrinsic outlines that "data have quality in their own right" [12] (p. 6), contextual underlines the dependence of data on its purpose. Simultaneously, representational and accessible depict, in a broader sense, the accessibility and usability of the data.

| USER-CENTRIC VIEW | | GENERALIST VIEW | | |
|---|---|---|---|---|
| **Intrinsic** | **Contextual** | **Inherent** ←→ | **System dependent** | |
| Believability<br>Accuracy<br>Objectivity<br>Reputation | Value-added<br>Relevancy<br>Timeliness<br>Completeness<br>Appropriate data amount | Accuracy<br>Completeness<br>Consistency<br>Credibility<br>Currentness | Accessibility<br>Compliance<br>Confidentiality<br>Efficiency<br>Precision<br>Traceability<br>Understand-<br>ability | Availability<br>Portability<br>Recoverability |
| **Representational** | **Accessibility** | | | |
| Interpretability<br>Ease of understanding<br>Representational consistency<br>Concise representation | Accessibility<br>Access security | | | |
| (a) | | (b) | | |

**Figure 2.** Data Quality dimensions from a (**a**) user-centric [12] and (**b**) generalist view [33], own illustration.

The concept of data quality dimensions is still applied today—albeit in differing approaches and scopes. For instance, in a recent literature review, Haug [34] discussed the existence of over 300 data quality dimensions with unique denotation; although, some referred either to identical or similar dimensions, where some equal dimensions were labeled divergently. This diversity highlights that no uniform definitions are utilized for data quality dimensions to date. Therefore, people utilizing data quality dimensions should rely on the most common definition and additionally provide a description.

The International Organization for Standardization also defines a rather technical set of data quality characteristics as part of standards for software quality within their data quality model in ISO/IEC 25012:2008(E) [33], distinguishing between two overarching data quality categories: inherent and system-dependent data quality (Figure 2). Here, inherent characteristics represent those with an intrinsic potential to meet the requirements, whereas the system-dependent characteristics suggest those conditional on the IT system used to manage the data. Here we presume, for example, that relevancy and timeliness of the data take on a different significance to the user (user-centric view) than to the software developer (generalist view), for whom availability and understandability may be more critical. Therefore, the views offer different perspectives on data quality, as the requirements among stake-holders may vary, as exemplarily presented in Figure 2. Regardless, from our point of view, the definitions of the dimensions themselves do not deviate with the view or perspective, supported by the given definitions below.

Here, we present a selection of nine frequently used data quality dimensions, focusing on the most utilized dimensions identified as intersection from literature reviews to cover the bigger picture. The reviews applied various methodologies, including content [35] and latent semantic analysis [36], and two reviews from the (medical) literature [37,38]. The data quality dimensions may apply to different data levels. In general, Haug [34] proposes six levels based on [39–42]: data item (a stand-alone value), data record (a row of values), data field (a column of values), data set (a table of values), database and database collection. However, the individual dimensions may not be reasonably applicable at all levels.

- Accessibility generally refers to data availability. In detail, the dimension depicts the extent to which the data are available and with which ease the user can access them [12,41,43]. The ISO standard ISO/IEC 25012:2008(E) [33] in particular measures accessibility considering the targeted use of the data and considers the absence of barriers, promoting the data accessibility for people with disabilities. No quantitative measure is suggested for measuring accessibility; instead, this dimension should be

assessed qualitatively or by a grade [44]. For example, we suggest utilizing a checklist that addresses documentation and machine readability issues.

- Accuracy is used to measure the degree of agreement between data and real-life object; it represents the extent to which the data are free of error and reliable [12] as well as the degree to which the data map (or are close to) the true values of the items [33,43], also understood as the correct value [45]. Accuracy may be calculated by dividing the accurate items or records by the total count of items or records [46]. As example, a local population register contains 932,904 phone numbers, and 813,942 have been confirmed, resulting in 87.25% (813,942/932,904 * 100) accuracy.

- Completeness is one of the most commonly applied data quality dimensions and generally provides a measure for the presence of data. The dimension maps the extent to which all expected and required data (records, attributes) are present or not missing, respectively [33]. Furthermore, it depicts the degree of sufficient resolution for the intended use [12,41]. Completeness may be computed by dividing the available items or records by the expected total number [46], resulting in a percentage if multiplied by 100. For example, a local population register contains all 932,904 inhabitants, but the date of birth is only available for 930,611 persons. This results in 99.75% (930,611/932,904 * 100) completeness.

- Consistency is also known as consistent representation and is the degree to which data are free of contradiction with themselves [33] or follow an established rule [43], and are provided in the same format [41], compatible with previous data [12]. Consistency may be represented as the proportion of items or records found to be consistent [46]. For instance, we presume the date of birth within a population register should be stored in the "YYYY-MM-DD" (year-month-day) format. In 61,196 of 930,611 total instances, date of birth was stored inverted as "DD-MM-YYYY", resulting in 93.42% ((930,611 − 61,196)/930,611 * 100) consistency.

- Currency or Currentness refers to the extent to which the data are sufficiently or reasonably up to date for the intended task [33,43]. The currency of data may be assessed qualitatively [44]. For example, a dataset of bird observations from the summer of 1969 is not suited for estimating bird populations in 2022. Otherwise, the share of current records within a population register may be determined by dividing the count of recently validated entries (764,111) by the total population count (932,904), which results in 81.91% (764,111/932,904 * 100) currency.

- Relevancy or Relevance as dimension maps the degree to which the data meets the expectations and requirements of the user [43,45]. Similar to currency, relevancy may be evaluated qualitatively at the discretion and in regard to the requirements of the user [44], e.g., by using a scorecard.

- Reliability is, in some instances, used as a synonym for accuracy [45]; despite that, others define the dimension as the extent to which an initial data value is on par with a subsequent data value [43]. This results in an identical quantification.

- Timeliness is also frequently applied and may depict the extent to which the age of the data is suitable for the intended use [12,41], or the time difference between a real-time event and the time the corresponding data capture or verification take [43,45]. Timeliness may be measured as duration [46], explicitly the time difference between data collection and entry. For example, if employees of a population register enter addresses into the database collected nine days before, the timeliness of this data is nine days.

- Validity of data maps the degree to which the data agree with established rules [43], resulting in a similar quantification as accuracy [46].

### 2.2.3. Data Quality Assessment

The individual data quality dimensions should be considered collectively as part of a comprehensive data quality assessment (DQA). The DQA process may consist of multiple steps, as suggested by DAMA UK [46] and visualized in Figure 3.

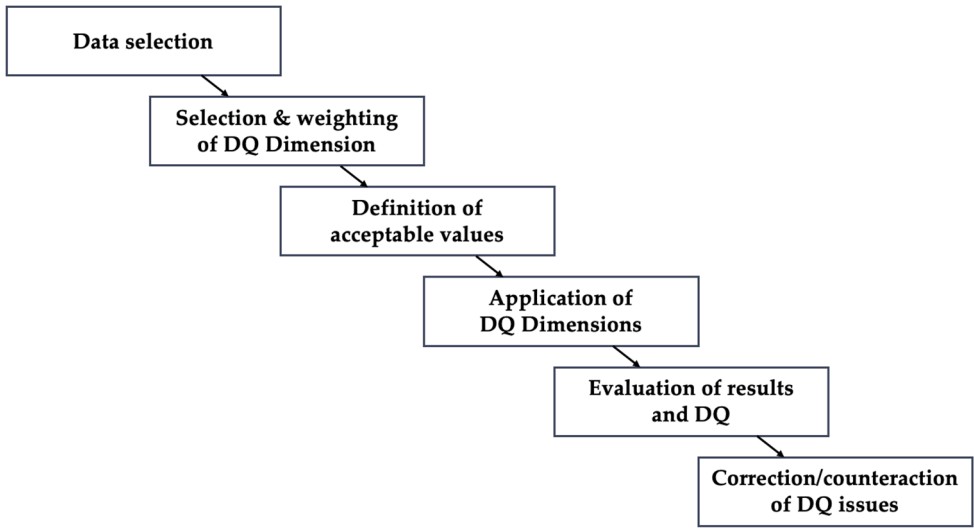

**Figure 3.** Data Quality (DQ) assessment steps suggested by DAMA UK [46], own illustration.

In the following, we illustrate the DQA steps presented in Figure 2 with an example. Let us assume we are conducting a survey pilot study, and now we want to evaluate the collected data. First, we determine which data we want to evaluate in terms of data quality: the entire questionnaire data, sections of the questionnaire, or single variables. In the next step, we determine which dimensions should be applied, e.g., completeness, consistency, and relevancy may be of high importance, as these dimensions enable us to determine the suitability of the questionnaire. Furthermore, we may weigh the selected dimensions in terms of their relevance. As mentioned in Section 2.2.2, the strategic importance of the individual data quality dimensions may depend on the organization, its users, the project, or other requirements. In the next step, the user should define the permitted value ranges and determine the scores for each considered dimension to identify suspicious scores or data, respectively (also see [47]). For instance, an acceptable value range of completeness for all individual variables may be 95%, which corresponds to a missing proportion of 5%. We then apply all selected dimensions to our data, enabling us to understand the data quality by the joint observation of all dimensions. If specific dimensions perform below expectations, we may implement measures to either correct the data errors (e.g., by using imputation methods for missing values) or prevent them in future surveys (e.g., by using input checks for digital questionnaires). Since we had conducted a pilot study, we should evaluate the data quality once more during or after data collection, converging to a cyclical process.

Zwirner [48] suggests three different approaches to address data issues with varying rigor, which may depend on the data collection type (one-time or continuous) or the general scale and importance of the project: In the laissez-faire approach, the user accepts present errors as either noncritical or too small/scarce to cause a notable effect on overall data quality. Therefore, the user undertakes only minor or no actions during the data cleansing process. The reactive approach comprises a singular, temporary act for data concerns, specifically eliminating a particular issue as it appears or is detected. However, the user implements no action to avoid or mitigate the problem in the future; the strategy relies on the repetition of the taken measure upon redetection of the error. The proactive approach is the comprehensive response to data quality problems: First, the user identifies and corrects data errors and addresses their sources to counteract subsequent flaws in advance. Zwirner also expects a genuine data quality management to include regular data quality and error monitoring.

With the potentially varying importance of individual data quality dimensions depending on the user or the intended data use, the need for a suitable and ideally tailored framework for data quality evaluation emerges, which organizes and controls the necessary

steps of data quality assurance. Moreover, the frameworks may promote transparency and comprehensibility of the data cleansing process. The literature suggests numerous general and special purpose frameworks for DQA and subsequent data improvement [35,49], including a framework for open data [50]. For example, in health research, Schmidt et al. [51] propose a modern framework for DQA, including practical software solutions in R. Thereby, the frameworks focus, at most, on objective criteria utilizing data quality dimensions.

### 3. Data Quality in Context and Practice

#### 3.1. Potential Consequences of Data Quality Issues

It appears plausible that wherever data issues exist, they may also have a negative impact. More than 20 years ago, and thus before the widespread and intensive use of information technology, people acknowledged that data issues were bad for business [52]. More recently, the negative effect of data issues in the United States was estimated to cost businesses $3 trillion (USD) per year [53], which can be explained in particular by the coping efforts by, e.g., scientists, decision-makers, and managers. The reinforcing problem is that data users often correct the data themselves without communicating the problems and requirements to data creators [53]; thus, the data correction work becomes an ever-recurring phenomenon. In addition to the additional efforts, Fürber [14] lists three other consequences in the organizational context that result from poor data: bad decisions, failed business processes, and failed projects. The consequences are not limited to higher costs or missed revenues but may also lower the quality of the offered products or services and cause dissatisfaction of other stakeholders [14].

Besides the potential impact on businesses and organizations, data issues may affect the practical work with data, e.g., as reported in population forecasting. Here, problems in data collection lead to errors in migration data, which in turn lead to errors in population estimates [54].

Just as population estimates, other projections also rely on data and therefore also on their quality. For instance, the simulations estimating the spread of SARS-CoV-2, the virus causing COVID-19, are modeled utilizing various data [55,56]. Particularly in the early stages of the pandemic, these types of simulations were discussed in Germany with great publicity and adduced by political decision-makers to impose protective measures like contact restrictions, temporary business closures, or curfews. However, criticism [57] points to a flawed database, which misled the simulations and thus characterized the subsequent containment measures as misguided, accompanied by significant consequences for civil liberties and the economy.

#### 3.2. What Affects Data Quality: The Case of Data Collection in Health Sciences

We now discuss factors related to data quality for research, using data collection in the health sciences as an example. Here, the discussion often focuses on bias, which may be defined as a systematic error in sampling or testing resulting from an (un-)intentional preference for a certain outcome or response over others [58]. Here, the accuracy dimension is of particular importance, namely when the recorded value differs from the actual one.

Research has indicated that splitting long web questionnaires increases the proportion of precise answers (e.g., reduces "neither/nor" answers) and leads to more detailed text in open questions, thus, increasing accuracy. However, when the response rate of the split questionnaires is compared with that of the original questionnaire, the response rate cannot be maintained and performs cumulatively worse [59]. In case of item or unit nonresponse, which is the nonresponse to a single question or a section of questions, the completeness dimension is negatively affected.

Other than questionnaire length, questionnaire mode potentially has implications on the data (quality). Research makes use of a wide range of survey methods, as examples: From face-to-face or telephone and computer-assisted interviews to self-administered questionnaires on paper or a (tablet-) computer. Bowling [60] provides six explanations on how data quality may be affected by the data collection mode:

1.  Impersonality: On the one hand, self-administered questionnaires may be preferred to inquire sensitive information, potentially triggering a sense of shame and leading to deviations from the true answer in interview situations (accuracy, consistency). On the other hand, the participant might be encouraged to respond and provide answers in all conscience (completeness, accuracy).
2.  Cognitive burden: The different methods require different efforts from the participant, such as listening, reading, comprehending, and classifying the answer. For example, during an interview, the participant usually has to listen, comprehend the question, and answer in natural language, while self-administered questionnaires require reading and writing or selecting pre-defined answers (completeness, accuracy).
3.  Study legitimacy: For phone interview participants, it may be difficult to judge the reputability of the study, leading to cautious answers or even non-response (accuracy, completeness, relevancy). In more official environments like a university's web page or on-site, increasing trust and confidence in participation may be expected.
4.  Questionnaire control: In interviews, the interviewer can guide the participant through the questions (accuracy, relevancy, completeness). In contrast, paper-based questionnaires offer little to no control of the question sequence at the participant's discretion; if applicable, written guidance is provided.
5.  Rapport: In face-to-face interviews, visual contact may simplify rapport establishment between interviewer and respondent compared to telephone interviews or even self-administered questionnaires, potentially increasing the motivation to participate (completeness) and mitigating the social desirability bias (accuracy).
6.  Communication: An interviewer can potentially elicit more information from the respondent than technically possible in self-administered questionnaires, for example, by further questioning or clarifying given answers (relevancy, accuracy, validity). Yet, social desirability bias may increase depending on the communication style (accuracy).

Apart from questionnaire mode, the layout and wording of queries also have implications for data collection, as both potentially influence the estimated prevalence of a target disease and associated risk factor assessment [61]. Therefore, we see the link to the accuracy and consistency dimensions. Furthermore, there is evidence for an interplay between questionnaire mode and wording [62]. These aspects of a rather technical nature are widely known and typically considered in modern questionnaire development, but should not be neglected due to their impact on data quality.

### 3.3. Challenges to Data Quality

Several data problems may arise along the data life cycle, exemplary presented in the following.

Data retrieval may be challenging during the data reuse phase due to the lack of directories or repositories and limited access options [32]. Furthermore, especially in the case of open data, the exact data origin or the data generation procedure may be ambiguous or inadequately documented, including insufficient or incomprehensive metadata [50], which may negatively affect data integration and analysis. Practical examples of data quality issues from an open data dataset include multiple occurrences of an individual, discrepancies in date formatting, different spellings of a condition, or missing values [63].

Then, poorly prepared data for reuse may complicate data linkage and processing; this especially applies to promptly and broadly available data from the WWW, potentially bringing along insufficient data (quality) control involving poor data structure, inaccuracies, compatibility issues, or missing data [63]. For example, a poor data structure may comprise data tables in popular spreadsheet formats utilizing color schemes instead of additional columns to represent data, posing a barrier for machine readability. More generally, data provision in highly compatible and machine-readable formats such as CSV is recommended over program-specific formats such as XLS (Microsoft Excel), as compatibility issues or unwanted data type conversion may arise while working with the data. For instance, date format options within Microsoft Excel may unknowingly alter the data, as reported

for altered gene names in bioinformatics [64,65]. Furthermore, a recent example of poor data reporting of data from the analysis perspective has been the periodical reporting of age-specific epidemiological surveillance data on COVID-19, which has been provided in part only in pdf files, especially in the early phases of the pandemic. This makes automated routines difficult, as the data has to be retrieved and structured manually first before being able to implement them in statistical analyses [66]. Furthermore, special care should be given to automatically generated data using scripts or algorithms without explicit information about their generation, as incorrectly implemented or faulty algorithms would result in incorrect data with implications for analysis. Likewise, especially for data obtained from the WWW, the user must pay particular attention to the origin and integrity since a low threshold for manipulation or sabotage may apply [67,68].

Subsequently, hardware compatibility or software problems may occur during the analysis phase, and most importantly, erroneous data can affect the analysis.

Next, in conceptualizing a project, lack of planning and expertise, including a poorly designed data (quality) management plan, may lead to preventable data problems during the data collection and quality assurance phase, leading to erroneous data collection or the inability to detect flawed data. Additional potential flaws may arise during data archiving, such as inadequate data description, including insufficient metadata provision, complicating data reuse. Also, incorrectly performed data archiving may result in data loss.

Finally, data generated for a specific usage is potentially helpful for additional purposes or questions; its reuse is part of the data life cycle completion. Here, difficulties may arise, as the data may not be entirely suited for its alienated use, for example, because the variables of interest may be too approximative (e.g., weekly vs. daily measurements) or missing to a high degree (which impacts analysis) [69].

Along the entire data life cycle, sufficient data integrity should be scrutinized before data use unless one is confident of preceding data checks or clearance. Applying the FAIR guiding principles (see Section 2.2.1), some of the stated problems can be addressed and mitigated.

## 4. Conclusions

Our contribution illustrated the importance of data quality for scientific research, responsibly informing the broad public, or guiding policies or businesses evidence-based. However, although we see data quality discussed as a vital key issue, it is even more astonishing that there are neither uniform definitions nor established frameworks to date, which could direct users through the process of data quality assessment and cleansing. Thus, standardization and transparency of the process are not yet achieved. We give a potential explanation in our work, namely that the data requirements are merely diverse depending on, among others, the project and the subjective factors associated with the individuals assessing the quality of the data. Here, dynamic and flexible data quality frameworks may become more relevant, though we see more of an opportunity for establishing frameworks depending on the data type, such as electronic health records.

We encourage our colleagues and the public to approach this complex topic to improve their data and recommend making their data quality considerations transparent and consistent, even without a unified framework. From our perspective and in the light of our case study presented in Section 3.2, we consider the data quality dimensions accuracy and completeness belong to the most critical data quality characteristics during data quality assessment, which is the crucial step to obtain knowledge concerning data applicability and usability. A high degree of accuracy may be achieved, for instance, by precise planning of data generation to ensure that the collected data reflect the actual value as accurately as possible. Random checks may also be undertaken after data generation to determine accuracy, depending on the data collected. In the end, better data could provide scientific research with improved opportunities for sophisticated and accurate statistical analyses, enriching our understanding of the underlying research problems. In particular, the current COVID-19 pandemic has shown us the value of well-structured, timely, and reliable

information that enables quick scientific analysis to allow for ad-hoc decision-making by governments. Previously well-defined and homogeneous data quality frameworks could facilitate the mentioned chain of action.

**Author Contributions:** Conceptualization, M.J.H. and P.V.; investigation, M.J.H.; writing—original draft preparation, M.J.H.; writing—review and editing, M.J.H.; visualization, M.J.H.; supervision, P.V.; project administration, M.J.H.; All authors have read and agreed to the published version of the manuscript.

**Funding:** This research received no external funding.

**Institutional Review Board Statement:** Not applicable.

**Acknowledgments:** M.J.H. receives a scholarship from the Life Science Foundation to promote science and research. We appreciate the helpful and timely remarks by the four anonymous reviewers of the paper.

**Conflicts of Interest:** The authors declare no conflict of interest.

**Entry Link on the Encyclopedia Platform:** https://encyclopedia.pub/20609.

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
