# Peer review of "Data Quality—Concepts and Problems"

_encyclopedia, doi:10.3390/encyclopedia2010032_

Round 1

Reviewer 1 Report

Dear authors, I am glad to read and review your manuscript, thank you for your research and contribution to the field.

Introduction: "Yet, the terms data and information are often used synonymously." Do the authors know the mathematical definition of Information from Shannon (information theory)? This might add more contrats to the differences / similarities between the terms data and information.

The manuscript is very well discussed and the arguments timely well presented. It is very comprehensive and yet reader friendly.

It is very itneresting to know how data quality concepts are defined depending on each field of research. This review summarizes some of them and the only missing aspect is a final recommendation from authors regarding to each steps woul dbe, in their opinions, the most important to follow.

Sehr gute Arbeit, hezrlichen Glückwunsch!

Author Response

Dear Reviewer,

We sincerely thank you for taking your valuable time to review and evaluate our submission concerning the topic of data quality. We gladly implemented your feedback to the best of our knowledge and belief; your input allowed us to refine the manuscript. All changes were made using the 'track change' (markup: all) option in MS Word, so all changes should be visible.

Kind regards.

Reviewer 2 Report

Brief Summary

I thank MDPI for offering me this opportunity to review this manuscript. In this work the authors addressed the data quality: concept and problem. The authors started by describing the history of recording of data, the writing is quite accessible and I enjoy reading it. In addition, the references are quite useful for readers to assess more background information about the data quality. However, while the current version does meet the aims and scopes for the encyclopedia where authors provide many reference information not only for researchers but a general public interested in accurate and advanced knowledge on data quality, current version may require more detail examples or concise description that focuses on addressing the data quality problem. Below are my comments, I hope author can find them useful.

General Comment

point: the author cite a reference Zhang et al. 2019 which provides more comprehensive version of the data quality problem. If possible authors may consider to create a section that describe those problems using a basic example, so reader can have a straightforward understanding. And I think that would make the contents fits the title.

point: Figure 1: the authors regraph and rephrase the Figure reference [18] and cite the work in the caption, it would be better to explain more how the cycle can be understood in general. For instance, for the red boxes (retrieval, integration and analysis), the authors may help reader to have a clear understanding on how the would related to the data quality problem.

point: Similarly to Figure 1, for the Figure 2 author directly use it from the references [14] with new arrangement, and summarize the reference [28]. If author stress on the data quality problem, author may help to clarify how those views are distinguishable from their interpretations. So the contribution of this work can be more clear.

point: In the section 3, the authors provide a case of data collection in health science, it is very comprehensive for a general reader to under stand the data quality (accuracy, consistency, relevancy validity). If possible, authors can help reader by provide another case (simpler example can be good) for the data quality, so a general public reader can quickly understand and learn the data quality issue.

point: Overall, authors provide sufficient background information in the reference. It would be great if authors can consider to provide more descriptions and help reader to understand the the data quality dimension issue (e.g. How the Figure 3 can be apply to another case study ?).

Author Response

(The authors gave the same response as above.)

Reviewer 3 Report

  • It is strongly recommended the authors restructure the paper improving its readability. For example the authors could start by defining the focus and scope of data quality in this document in the introduction, briefly stating what it denotes and then providing the contents of the other sections. Then they could continue with the history etc., definitions, dimensions, approaches, challenges, conclusion.
  • Considering that data collection is a modern alternative of collecting observations, making a connection to the scientific method and the original ancient Greek word for quality could support defining better "data quality" in the document. Furthermore, please consider discussing the terms "data quality" in a complementary and gradual approach in the document, such as first the quality (of observations/logs/other), the term data, and then data quality.
  • There are several statements throughout the document that are not supported by references, thus could be conceived as subjective and/or misleading. To avoid this, please add the corresponding references in every statement that is not directly deduced by the previous sentence(s), such as: 
    • " centralized data centers gathered data and were business-orientated"
    • " Since 2000 and with the increasing spread of the internet, data centers have expanded their capacities to cloud computing"
    • " Furthermore, specific scientific disciplines, such as epidemiology, acquire data to research health conditions and their causes"
  • In the title of the 2nd section "Approaches to capture Data Quality ", the meaning, especially of the word capture in this context, is not clear. Please consider rephrasing.
  • I think mentioning as part of the data quality the different dimensions and then the different ways to measure and quantify the quality of data could improve the readability of the document.
  • In subsections 2.1 and 2.2 the terms "data lifecycle" and "data quality lifecycle" are used, yet their meaning and/or differences are not clear. Are they used for the same purpose? Furthermore, it is not clear how data quality and data lifecycle are related and how the lifecycle of data affects their quality. Please elaborate.
  • In subsection 2.2, "open data" are mentioned, yet there is no reference or explanation to them. Please update accordingly. Moreover, how open data are related to the quality of data? Please explain.
  • In subsection 2.2 and regarding the following "More generally, data provision in universal formats such as CSV is recommended over program-specific formats such as XLS (Microsoft Excel) or SAV (IBM SPSS), as compatibility issues or unwanted data type conversion may arise while loading the data", it is not clear why format conversion is considered an issue. An example could help. Also, please add the related references about the recommendation, program-specific and universal formats and the conversion issues.
  • FAIR principles are discussed at the end of 2.2 and 3.1. Are FAIR principles related to the quality of data? If yes, how? Please elaborate using adequate references to support the outcomes of the discussion.
  • The authors state that the document discusses on the importance of data quality for the scientific research. Yet, the document seems to be focused on health sciences, rather than mathematics or computer science. There is no reference to the importance of a proper design of experiment to support the scientific method or the skills for collecting, managing, processing, and interpreting collected data. The authors should expand their definition of data quality to capture all these aspects as well, in order to provide the reader/researcher with a proper understanding of data quality and its importance. It is strongly suggested to add more information and works about the background of data quality, its dimensions, and maybe difference to data protection. Brief statements with the corresponding references and some discussion on the outcomes would be sufficient.
  • Moreover, discussing on additional use cases from other domains would benefit the paper.  

Author Response

(The authors gave the same response as above.)

Reviewer 4 Report

In my opinion, the paper includes the most important issues related to data quality. However, there are a few issues that might be developed. I suggest highlighting the importance of data quality in different industries. For instance, poor data quality of evaluating the success of a new product can lead to business failure. Moreover, the issues related to the improvement of data quality might be presented.

Author Response

(The authors gave the same response as above.)

Round 2

Reviewer 1 Report

Authors have adressed the comments, the final version is, in my opinion, ready to be published.

Author Response

Dear Reviewers,

Again, thank you for your time and efforts to consider our answers and the revised version of the manuscript. We are glad that you found your constructive feedback realized. We felt encouraged to reconsider the paper's structure critically and rearranged some paragraphs to improve readability.

In detail, we rearranged Section 1 following Reviewer 3's earlier suggestion, starting with the definition of the terms data and information, followed by a short historical context, after which we give the definition of quality, followed by the description of the term data quality. Moreover, we added a closing paragraph to Section 1, which provides a short overview of the structure of the paper. Beyond that, we followed the suggestion of putting the section on data quality challenges as a separate section (now Section 3.3.) before the conclusions in Section 4.

Kindest regards.

Reviewer 2 Report

I thank authors to take my comments and give their responses accordingly. 
For this revision, I think the authors have addressed my comments. Moreover, after reading the authors' responses to other reviewers' comments and the resubmitted manuscript, I think the authors have provided a better version of ms when comparing with the ms in their first submission. 
However, after reading carefully for the revised version,  I would agree with a reviewer who concerns about the readability for the authors' first version and the structure of the paper. 
So if authors can be more serious to restructure their revised version, it would be a more accessible paper and helps  readers to gain the knowledge of data quality easily.
However, I would also respect the authors' argument that they would like to retain the current structure (the authors provide several references to back up their structure).

Author Response

(The authors gave the same response as above.)

Reviewer 4 Report

I accept the revised version of the manuscript.

Author Response

(The authors gave the same response as above.)

Round 3

Reviewer 2 Report

I thanks the authors to take my comments and give their response in details. From my perspective, all of my comments are addressed, and I think the manuscript is ready to go.